# Cytogenetic Identification and Molecular Marker Analysis of Two Wheat–*Thinopyrum ponticum* Translocations with Stripe Rust Resistance

**DOI:** 10.3390/plants14010027

**Published:** 2024-12-25

**Authors:** Guotang Yang, Yi Han, Huihui Yin, Xingfeng Li, Honggang Wang, Yinguang Bao

**Affiliations:** 1State Key Laboratory of Wheat Improvement, Shandong Agricultural University, Tai’an 271018, China; yangguotang@sdau.edu.cn (G.Y.); 13235389658@163.com (Y.H.); lixf@sdau.edu.cn (X.L.); hgwang@sdau.edu.cn (H.W.); 2Agronomy College, Shandong Agricultural University, Tai’an 271018, China; 3Liaocheng Academy of Agricultural Sciences, Liaocheng 252000, China; huihuiyin84@163.com

**Keywords:** wheat, *Thinopyrum ponticum*, translocation line, stripe rust, cytogenetic analysis, intron-targeting markers

## Abstract

Stripe rust, induced by *Puccinia striiformis* f. sp. *tritici* (*Pst*), is one of the most destructive fungal diseases of wheat worldwide. *Thinopyrum ponticum*, a significant wild relative for wheat improvement, exhibits innate immunity to this disease. To transfer the stripe rust resistance gene from *Th. ponticum* to wheat, two translocation lines, SN21171 and SN52684, were produced through distant hybridization techniques. Disease evaluation results showed that these two lines were immune to *Pst* species CYR32 at the adult plant stage. Molecular cytogenetic analyses and specific intron-targeting markers amplification results revealed that SN21171 and SN52684 harbor several T3E^b^-3DS·3DL and T1E^b^-1BS·1BL translocation chromosomes. Furthermore, the comparison of the chromosome karyotype from two translocation lines and their recurrent parent YN15, revealed that structural variation occurred in chromosomes 2A, 5A, 2B, 4B, 5B, and 6B in SN21171 and chromosomes 5A, 3B, 4B, 5B, 6B, and 7B in SN52684. Agronomic trait assessments uncovered advantageous properties in both lines, with SN21171 matching the recurrent parent and SN52684 exhibiting elevated higher grain number per main spike and increased thousand grain weight. These two translocation lines and specific markers may apply to wheat stripe rust-resistance breeding.

## 1. Introduction

Wheat stripe rust, caused by *Puccinia striiformis* West. f. sp. *tritici* Eriks. & Henn. (*Pst*), is one of the most devastating wheat diseases and severely restricts wheat production worldwide. *Pst* species thrive in moisture and cool climates, which infect wheat leaf tissues from the one-leaf stage to maturity [1]. More recently, the stripe rust was endemic in non-traditional, warmer and dryer areas, indicating that new species exhibit greater variability and adaptability [2]. Yield losses due to stripe rust typically range from 0.1% to 5%, although severe outbreaks can result in losses of 5% to 25% [3]. For example, destructive epidemics caused yield losses exceeding 6.0, 3.2, 1.8, 1.3, and 1.5 million metric tons in 1950, 1964, 1990, 2002, and 2017 years in China, respectively [4,5,6,7]. Historically, breeding resistant varieties are considered to be an economical and environmentally friendly method to control stripe rust. Therefore, developing germplasm carrying novel resistance genes and incorporating them into breeding programs can mitigate wheat yield reductions.

Up to now, although more than 80 stripe rust resistance genes have been identified and cataloged, only 10 genes have been isolated [8]. Among these, *Yr5*, *Yr7*, *YrSP*, *YrAS2388R*, and *YrU1* encode nucleotide binding and leucine-rich repeat (NLR) proteins [9,10,11]. Additionally, two multi-pathogen (partial) resistance genes, *Yr18/Lr34/Sr57/Pm38* and *Yr46/Lr67/Sr55/Pm39*, encode an ATP-binding cassette transporter and hexose transporter, respectively [12,13]. *Yr15* features two kinase-like domains and *Yr36* possesses a kinase domain and a lipid binding domain, both conferring broad-spectrum resistance to stripe rust [14,15]. *YrNAM* includes a No Apical Meristem (NAM) domain and a BED Zinc Finger (ZnF-BED) domain [8]. The virulence pattern of *Pst* can rapidly evolve, potentially rendering resistant genes ineffective [16]. For example, the CYR29 species, virulent to *Yr9*, and the V26 species, virulent to *Yr24/Yr26*, were first reported in China in 1987 and 2008, respectively [17]. Thus, mining and cloning new types of *Yr* genes, and pyramiding multiple *Yr* genes is crucial to overcome the fast-changing *Pst* isolates. 

Wheat wild relatives harbor a wealth of valuable genes for enriching wheat genetic diversity. Some *Yr* genes have already been transferred from alien species to wheat using distant hybridization and chromosome engineering, such as *Yr83* from *Secale cereal* L. [18], *Yr5V* from *Haynaldia villosa* (L.) Schur [syn. *Dasypyrum villosum* (L.) Candargy] [19], *Yr50* from *Thinopyrum intermedium* (Host) Barkworth and D. R. Dewey [syn. = *Agropyron intermedium* (Host) Beauvoir = *Elytrigia intermedia* (Host) Nevski] [20], and *Yr69* from *Thinopyrum ponticum* (Podp.) Barkworth and D. R. Dewey [syn. = *Agropyron elongatum* (Host) P. Beauvois = *Lophopyron ponticum* (Podp.) Á Löve = *Elytrigia elongata* (Host) Nevski = *Elytrigia pontica* (Podp.) Holub.] [21]. Because *Th. ponticum* (2*n* = 10*x* =70; E^e^E^e^E^b^E^b^E^x^E^x^StStStSt or JJJJJJJ^S^J^S^J^S^J^S^) is a decaploid species with a complex and large genome, additional *Yr* genes will need to be identified and functionally annotated in future studies except for the permanently documented gene *Yr69*.

With the development of cytogenetic methods for identifying the alien segments, certain wheat–*Th. ponticum* addition and substitution lines have been developed by using chromosome engineering. For example, wheat–*Th. ponticum* addition line WTA55, derived from partial amphiploid Xiaoyan 7430 and carried one pair of alien group-6 chromosomes, has demonstrated resistance to stripe rust species at both the seedling and adult plant stages [22]. Wheat–*Th. ponticum* substitution lines, ES-9 and ES-10, separately carried 2St and 3St chromosomes and showed stripe rust resistance at the adult plant stage and all stages [23]. As is well-known, the translocation lines carry shorter alien segments and lesser genetic linkage drags, representing the most desirable germplasms for wheat genetic improvement. However, there are limited reports on the development of wheat–*Th. ponticum* translocation lines with stripe rust resistance.

Our group has focus on the production and application of wheat–*Th. ponticum* translocation lines with excellent disease resistance and good agronomic performance. In this study, two wheat–*Th. ponticum* translocation lines with stripe rust resistance were created by distant hybridization. The chromosome constitutions were elucidated by molecular cytogenetic methods and intron-targeting marker amplification. These selected intron-targeting markers could trace the alien segments rapidly and accelerate the transfer of resistance. In addition, their agronomic traits were evaluated. This study provides an important foundation for the breeding application of stripe rust-resistant germplasms.

## 2. Materials and Methods

### 2.1. Plant Materials

The wheat–*Th. ponticum* translocation lines SN21171 and SN52684 were progenies of *Th. ponticum* and wheat cultivar Yannong 15 (YN15). The SN21171 and SN52684 were developed as: YN15//*Th. ponticum*/YN15. *Th. ponticum* (accession No. R431) was provided by Prof. Zhensheng Li, formerly of the Northwest Institute of Botany, the Chinese Academy of Sciences, Yangling, China. The wheat variety Huixianhong (HXH) served as the susceptible control and was used in stripe rust resistance evaluation. HXH and YN15 were preserved at our laboratory, College of Agronomy, Shandong Agricultural University.

### 2.2. Stripe Rust Resistance Evaluation

At the adult stage, resistance to stripe rust was evaluated after natural infection in field-grown plants at the Experimental Station of Shandong Agricultural University over three growing seasons (2020–2023). The most severe reaction type in 2022 was considered to be the final resistance results. Twenty-five plants were grown in 1.5 m long rows, with 25 cm between the rows. HXH was planted perpendicularly next to the test rows as an inoculum spreader and a susceptible control. The disease symptoms were recorded at the grain-filling stage. The infection types (ITs) of stripe rust were scored using a 0–4 scale [24]. In detail, “0” represents immunity, with no visible uredinia and necrosis on leaves; “;” represents near immunity, with no visible uredinia and hypersensitive flecks on leaves; “1” represents high resistance, with small uredinia surrounded by necrosis; “2” represents moderate resistance, with small to medium uredinia surrounded by necrosis; “3” represents moderate susceptibility, with medium uredinia without chlorosis or necrosis; and “4” represents high susceptibility, large uredinia without chlorosis or necrosis. Plants with IT scores 0–2 were regarded as resistant, whereas those with scores of 3–4 were indicated to be susceptible.

### 2.3. Cytogenetic Analyses

According to [25], the methods of sequential GISH and FISH required the least modification. Briefly, the metaphase chromosomes of root tips cells of these translocations were prepared. The total genomic DNA (gDNA) of *Th. ponticum* was labeled with Texas-red-5-dCTP (red) and used as a probe, while the gDNA of YN15 was used as a block. The ratio of the probe to blocker was 1:200. The slides were counterstained with 4, 6-diamidino-2-phenylindole (DAPI).

For the same cells, eight single-strand oligonucleotides (oligos), including TAMRA (6-carboxytetramethylrhodamine)-labeled oligos pAs1-1, pAs1-3, pAs1-4, pAs1-6, AFA-3 and AFA-4, and FAM (6-carboxyfluorescein)-labeled oligos pSc119.2-1 and (GAA)_10_, were pooled to form modified multiplex#4 and used for mc-FISH analysis after the GISH analysis [26]. All probes were synthesized by Sangon Biotech Co., Ltd. (Shanghai, China). The 20 µL of hybridization buffer contained 7 µL of ddH2O, 7.5 µL of FA (formamide F9037, 99.5%, Sigma, Steinheim, Germany), 1.5 µL of 20× saline sodium citrate (SSC), 5 µL of salmon sperm DNA, 1 µL of bulked oligo probe, and 2.5 µL of 50% dextran sulfate (Cat#D0007, SunShineBio, Nanjing, China). The hybridization buffer for in situ hybridization was denatured at 100 °C for 13 min and then transferred to −20 °C 100% alcohol for at least 10 min. Chromosome preparations were denatured in 70% alcohol + NaOH (0.15 mol/L) at 44 °C for 5 min. Hybridizations were conducted at 37 °C incubator for more than 6 h. Slides were washed with 2× SSC at 44 °C for 10 min, washed with water, and then air-dried. The cells that have integrated chromosomes and good hybridization signals were selected and captured using Olympus DP80 CCD camera attached to an Olympus BX60 and merged by the program CellSens Standard 1.12 (Olympus, Tokyo, Japan).

### 2.4. Molecular Marker Analysis

The intron-targeting markers were developed based on the sequence conservation of orthologous genes. Therefore, they have higher versatility between closely related species. In this study, the intron-targeting markers specified for *Dasypyrum villosum* not only detect alien chromosomes but also determine its homoeologous groups in wheat–*Th. ponticum* translocation lines [27]. PCR amplification system was performed in a volume of 15 µL, containing 7.5 µL 2× Power Taq PCR MasterMix, 1 μL of template DNA (100 ng/μL), 1 μL of each primer (2.5 μmol/L), and 4.5 μL of ddH_2_O. The PCR reactions were as follows: one cycle at 94 °C for 2 min for denaturation; thirty cycles at 94 °C for 40 s, 50–65 °C (depending on annealing temperature for each marker) for 45 s, and 72 °C for 1 min; and one cycle at 72 °C for 5 min for final extension. The PCR products were separated into 8% non-denaturing polyacrylamide gels. The band patterns were visualized by silver staining and photographed with the Tanon Gis-2010 Gel Image System (Tanon, Shanghai, China).

### 2.5. Agronomic Traits Evaluation

During the growing season (2021–2022), wheat–*Th. ponticum* translocation lines SN21171 and SN52684, and backcross parent YN15 were all planted at the Experimental Station of Shandong Agricultural University. The agronomic traits, mainly including plant height, spike number per plant, main panicle length, grain number per main spike, and thousand-grain weight, were investigated and analyzed.

## 3. Results

### 3.1. Evaluation of the Resistance of Stripe Rust

The *Puccinia striiformis f. sp. tritici (Pst)* species CYR32 was used to evaluate the reaction of the translocation lines SN21171, SN52684, and their parents to stripe rust. The results showed that the susceptible control HXH showed high ITs (IT = 4), the translocation line SN21171 and *Th. ponticum* showed immunity (IT = 0) to stripe rust, the translocation line SN52684 showed high resistance to stripe rust (IT = 1), and recurrent parent YN15 showed moderate susceptibility to stripe rust (IT = 3) (Figure 1). The results indicated that the stripe rust resistance of the two translocations was derived from *Th. ponticum*.

### 3.2. Chromosome Constitution and Structural Variation

The GISH and mc-FISH analyses were performed to determine the chromosomal constitution of SN21171 and SN52684. GISH analysis revealed that two translocation lines both carry 40 wheat chromosomes and 2 wheat–*Th. ponticum* translocated chromosomes (Figure 2a,c). In the mc-FISH analysis of SN21171, stronger and clear pAs1 signals were located in the terminal regions of the short arm and long arm of translocation chromosomes, respectively. This result was similar to the wheat standard karyotype of chromosome 3D (Figure 2b). Thus, we speculated that chromosome 3D was broken in its short arm and the longer segment was fused to the alien chromosome segment to produce the novel translocated chromosome in SN21171. In the mc-FISH analysis of SN52684, stronger and clearer punctated pSc119.2 signals were detected in the terminal regions of the long arm and centromere zone of translocation chromosomes, respectively. These results indicate that the alien segments were translocated to the short arm of wheat chromosome 1B (Figure 2d). Additionally, FISH signals of the wheat chromosomes in SN21171 and SN52684 were compared with those in its recurrent parent YN15. Differences between SN21171 and YN15 were found in the middle regions of 2AL, 5AL, and 2BL, and the terminal regions of 5AS, 4BS, 4BL, 5BS, 6BS, and 6BL (Figure 3). Different fluorescence signals between SN52684 and YN15 were found in the middle regions of 5AL and the terminal regions of 5AS, 3BL, 4BS, 4BL, 5BS, 6BS, 7BS, and 7BL. These differences indicated that related regions underwent structural variations in the formation of SN21171 and SN52684.

### 3.3. Intron-Targeting Marker Analyses

To analyze the genome and homoeologous group attributions of alien segments in SN21171 and SN52684, a total of 841 intron-targeting markers were amplified in the gDNAs of *Th. ponticum*, *Th. elongatum*, *Th. bessarabicum*, *Pseudoroegneria*, SN21171, SN52684, and their wheat parent YN15. The amplification results showed that 30 intron-targeting markers belonging to the third homoeologous group were able to amplify specific bands in *Th. ponticum* and SN21171, which were absent in the wheat parent YN15. Furthermore, 3 of 30 intron-targeting markers could amplify the same bands in *Th. ponticum*, *Th. bessarabicum* and SN21171 genomes, but could not in the *Th. elongatum* and *Pseudoroegneria* genomes (Figure 4a–c, Appendix A). In the same way, six intron-targeting markers belonging to the first homoeologous group showed amplification with prominent bands specific to *Th. ponticum* in SN52684, which were absent in YN15. In addition, two of these six intron-targeting markers could amplify the same bands in *Th. ponticum*, *Th. bessarabicum*, and SN52684 genomes, but could not in *Th. elongatum* and *Pseudoroegneria* genomes (Figure 4d,e). As such, we confirmed that SN21171 and SN52684 separately carry T3E^b^-3DS·3DL and T1E^b^-1BS·1BL translocation.

### 3.4. Agronomic Performance of SN21171 and SN52684

The phenotypic characteristics of SN21171, SN52684, and their recurrent parent YN15 were investigated in the 2021–2022 period. Compared with YN15, SN21171 had similar plant height, spike number per plant, spikelet number per main spike, grain number per main spike, and thousand-grain weight. However, SN52684 had longer main panicles and higher grain number per main spike and thousand-grain weight than YN15 (Figure 5). The yield potential of both translocation lines is currently being determined in field trials.

### 3.5. Discussion

*Th. ponticum*, as a tertiary pool of common wheat, has many valuable genes useful for wheat genetic improvement. Transferring resistance genes from *Th. ponticum* to wheat was proven to be an effective approach for wheat improvement. Up to now, many germplasms with desirable traits have been created from the distant hybridization between common wheat and *Th. ponticum*. For example, the partial amphiploids of Xiaoyan series, such as Xiaoyan 693, Xiaoyan 784, Xiaoyan 7430, and Xiaoyan 7631, were highly resistant to stripe rust species CYR32 and CYR33 [28]. ES-10, whose wheat 3D chromosomes were replaced by a pair of *Th. ponticum* 3St chromosomes, was highly resistant to stripe rust at all stages [23]. Wheat cultivar Xiaoyan 6, produced by distant hybridization between wheat and *Th. ponticum*, exhibited high quality, multi-resistance, and good agronomic performance. More than 80 cultivars have been derived from this backbone parent and their accumulated extending area exceeds 300 million mu throughout China [29].

Wheat–*Th. ponticum* partial amphiploid Xiaoyan 7430 has good resistance to stripe rust. To transfer its resistance gene into the wheat background, new substitution and addition lines have been developed by distant hybridization. Among them, X005 was a 6J^s^ (6B) substitution line, whose 6J^s^ chromosomes were responsible for the stripe rust resistance using linkage analysis [30]. Addition line WTA55 with alien group-6 chromosomes produced by the cross Xiaoyan 7430 and common wheat Xiaoyan 81, which was evaluated as resistant to stripe rust species at both the seedling and adult plant stages [22]. Thus, alien group-6 chromosomes of Xiaoyan 7430 carry novel stripe rust resistance genes. *Yr69* from the wheat–*Th. ponticum* introgression line CH7086 was mapped on wheat chromosome 2AS. If it is a compensating translocation, *Yr69* may originate from the group-2 chromosome of *Th. ponticum* [21].

Previous studies confirmed that genome re-sequencing is useful for wheat genome-wide association studies (GWAS), population analysis, and other applications. For example, Hao et al. conducted a GWAS using 91 wheat cultivars and found that eight genes located in the starch synthesis pathway were strongly selected by breeding [29]. Niu et al. performed a comprehensive comparative analysis of a whole-genome resequencing panel of 355 common wheat accessions at the phenotypic and genomic levels and the results indicated that different genetic architectures of wheat from China and the United States aimed to adapt to diverse agricultural production environments [31]. In recent years, genome re-sequencing has been applied to study wild relatives of wheat. The stripe rust resistance gene *YrT14* was located in an 88.1 Mb region on *Th. intermedium* chromosome 19 based on the karyotype, and the reaction to stripe rust and the genome resequencing data of different wheat–*Th. intermedium* translocation lines [32]. Han et al. analyzed the Robertsonian translocations T1AL·1PS and T1AS·1PL and determined the chromosomal junction point based on the resequencing results [33]. In our study, depending on the intron-targeting marker amplification, the genome and the homoeologous groups of their alien segments were confirmed in wheat–*Th. ponticum* translocation lines SN21171 and SN52684, which also helped us to identify the origin of the stripe rust resistance gene of these translocation lines. Furthermore, we combined the genome re-sequencing data to determine the location of the translocation breakpoint.

Pedigree derivation has frequently been used to confirm the origin of resistance genes in wheat–alien introgression lines. For example, the wheat–*Th. ponticum* translocation line WTT34 was developed by crossing *Th. ponticum* with the wheat cultivars Misui, Zhengyou 7, 8602, Jimai 11, and Xiaoyan 81. At the seedling stage, the evaluation of stem rust reactions suggested that WTT34 is highly resistant (IT = 1;) to Ug99 species PTKST and *Th. ponticum* is nearly immune (IT = 0;) to PTKST, whereas the remaining wheat parents are susceptible to infection (ITs ranging from 3 to 3+). These results indicated that WTT34 possesses a new Ug99-resistance gene derived from *Th. ponticum* [34]. As another example, the wheat–rye 4R chromosome disomic addition line WR35 was created by crossing Xiaoyan 6 with the rye cultivar German White. At the adult stage, WR35 is highly resistant (IT = 1) to a mixture of *Bgt* isolates, German White is immune with 0 IT, while Xiaoyan 6 and controls all show susceptible reactions with 8–9 ITs. The powdery mildew resistance of WR35 is thus presumably derived from rye [35]. According to these results, for germplasm resources with known family trees, pedigree analysis is a convenient, feasible method for tracing the origin of disease-resistance genes. In this study, SN21171 and SN52684 were both developed by crossing *Th. ponticum* and YN15. SN21171 and its parent *Th. ponticum* were immune to stripe rust species in yield, SN52684 was highly resistant with IT 1, whereas YN15 showed susceptible symptoms. These results indicated the resistance genes of SN21171 and SN52684 were all derived from *Th. ponticum*. Furthermore, the alien segments of SN52684 and SN21171 were found in several 1E^b^ and 3E^b^ chromosomes of *Th. ponticum*, which suggested that the stripe rust resistance of SN52684 and SN21171 originates from novel genes derived from *Th. ponticum*, respectively.

As a vast reservoir of potentially valuable genes, wheat’s wild relatives have been used to enrich wheat genetic diversity through developing chromosome engineered germplasm. To date, many valuable translocation lines carrying desirable traits and little genetic linkage drag have been widely used in wheat breeding. For example, the wheat–rye 1RS·1BL translocation has contributed to producing approximately 30% of wheat varieties released after the year 2000 [36]. More than 40 wheat cultivars carrying the wheat–*Dasypyrum villosum* 6VS·6AL translocated chromosomes have been released and cultivated on a large scale in China [37]. In this study, Compared with YN15, SN21171 had similar plant height, spike number per plant, spikelet number per main spike, grain number per main spike, and thousand-grain weight. The grain number per main spike of SN52684 was more than YN15 and the thousand-grain weight of SN52684 was heavier than YN15. Thus, the SN21171 and SN52684 carried excellent stripe rust resistance and good agriculture traits, making them potentially useful for breeding new disease-resistant wheat cultivars.

## 4. Conclusions

Two wheat–*Th. ponticum* translocation lines, SN21171 and SN52684, with desirable stripe rust resistance were developed by distant hybridization. Cytogenetic analyses and intron-targeting marker amplification indicated that SN21171 carries a T3E^b^-3DS·3DL translocation and SN52684 underwent a T1E^b^-1BS·1BL translocation event. Furthermore, agronomic trait assessment showed that SN21171 was similar to YN15 and SN52684 had increased grain numbers per main spike and thousand-grain weight. Due to good stripe rust resistance and agronomic performance, both translocation lines will contribute to wheat breeding for disease resistance.

## Figures and Tables

**Figure 1 plants-14-00027-f001:**
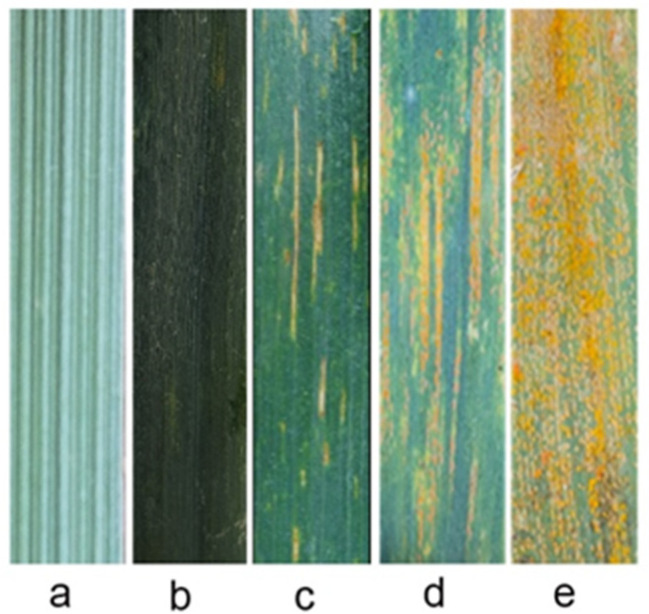
The disease resistance evaluation of SN21171, SN52684, and their parent YN15 at the adult plant stage. (**a**) *Th. ponticum*; (**b**) SN21171; (**c**) SN52684; (**d**) YN15; (**e**) HXH.

**Figure 2 plants-14-00027-f002:**
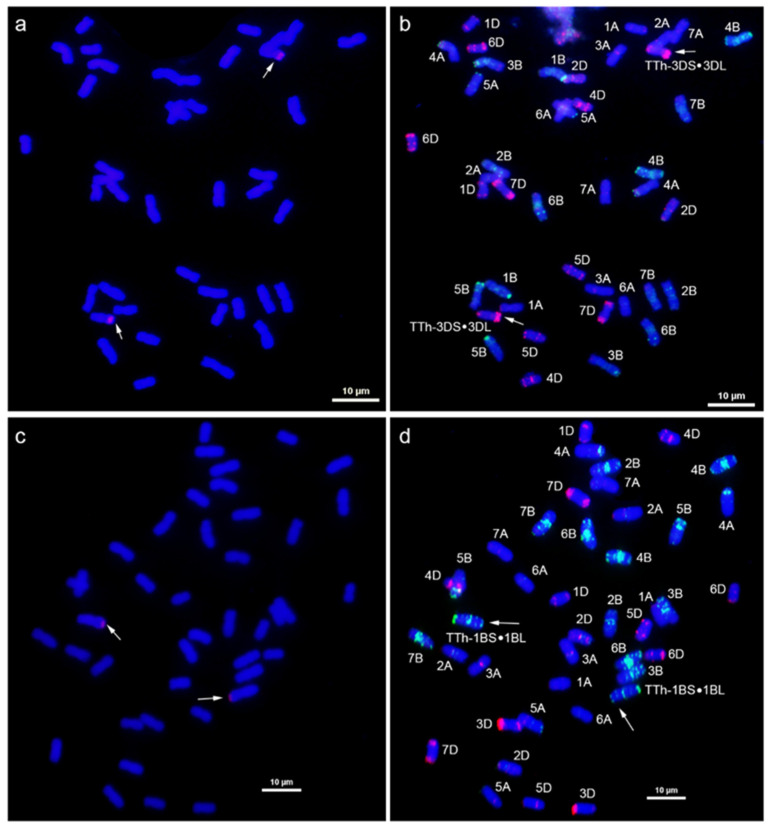
Cytogenetic analyses of SN21171 and SN52684. The GISH patterns of SN21171 (**a**) and SN52684 (**c**) with *Th. ponticum* gDNA as a probe. The mc-FISH patterns of SN21171 (**b**) and SN52684 (**d**) using eight single-strand oligos. The arrows note a pair of translocated chromosomes. Bar = 10 μm.

**Figure 3 plants-14-00027-f003:**
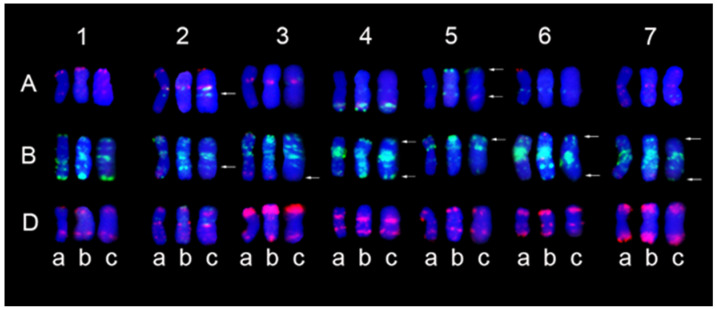
Comparison of chromosome karyotypes in SN21171 SN52684, and YN15. (**a**) YN15; (**b**) SN21171; (**c**) SN52684. The arrows note locations with different FISH bands.

**Figure 4 plants-14-00027-f004:**
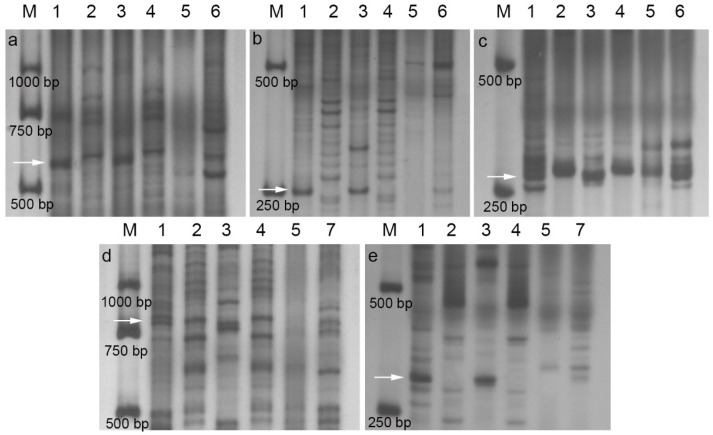
Specific intron-targeting marker amplification in SN21171 and SN52684. (**a**) CINAU1185; (**b**) CINAU1195; (**c**) CINAU1214; (**d**) CINAU957; (**e**) CINAU959. M: marker; 1: *Th. ponticum*; 2: *Th. elongatum*; 3: *Th. bessarabicum*; 4: *Pseudoroegneria*; 5: YN15; 6: SN21171; 7: SN52684. The arrows indicate specific bands.

**Figure 5 plants-14-00027-f005:**
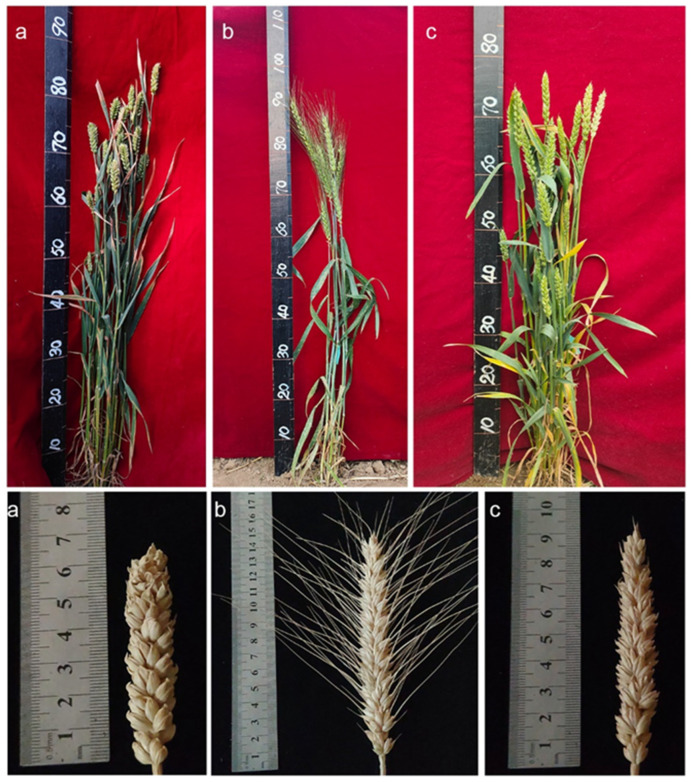
The agronomic performance of SN21171, SN52684, and YN15. (**a**) SN21171; (**b**) SN52684; (**c**) YN15.

## Data Availability

All data generated or analyzed during this study are included in this published article and its Appendix A.

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
