# Peer review of "Cytogenetic Identification and Molecular Marker Analysis of Two Wheat–Thinopyrum ponticum Translocations with Stripe Rust Resistance"

_plants, 2024, doi:10.3390/plants14010027_

Round 1

Reviewer 1 Report

Comments and Suggestions for Authors

The research presented in the manuscript (plants-3294401) was focused on production and application of wheat -Th. ponticum translocation lines with excellent disease resistance to stripe rust, caused by Puccinia striiformis f.sp. Tritici (Pst) and good agronomic performance. In this study two new wheat-Th. ponticum translocation lines with stripe rust resistance were created by distant hybridization. So far, there are only few reports on the development of wheat-Th. ponticum translocation lines resistant to this disease. Experimental methods used in this study (cytogenetic analyses-GISH and FISH; molecular marker analysis-Intro targeting marker amplification ) were chosen properly and allowed to obtain the results which indicated the resistance genes of SN21171 and SN 52684 were all derived from Th. ponticum. Besides, they suggested that stripe rust resistance of both translocation lines originates from novel genes derived from Th. ponticum. So that, these lines will contribute to wheat breeding resistance. References are appropriate and conclusions are consistent with the evidence and arguments presented in this research. Regarding the significant scientific value of this study for wheat breeding I recommend to publish this manuscript in Plants (some editorial errors noticed in the text were already advisable  in the previous revision).

Author Response

Dear editor:

Thank you very much for sending us the reviewers’ comments on our manuscript (Plants-3294401) entitled “Cytogenetic identification and molecular marker analysis of two wheat-Thinopyrum ponticum translocations with stripe rust resistance”. Those comments are all valuable and helpful for revising and improving our paper. The following is the answers and revisions we have made in response to the reviewers’ questions and suggestions on an item by item basis. We have checked the paper again for English editing and the changes to the manuscript were also highlighted in yellow text.

Thanks again for your consideration of our manuscript. We are willing to respond to any further questions and comments that you may have.

Yours sincerely

Guotang Yang & Yinguang Bao

In response to the reviewers’ comments: (reviewers’ comments are marked in bold)

Q1: The research presented in the manuscript (plants-3294401) was focused on production and application of wheatTh. ponticum translocation lines with excellent disease resistance to stripe rust, caused by Puccinia striiformis f.sp. Tritici (Pst) and good agronomic performance. In this study two new wheatTh. ponticum translocation lines with stripe rust resistance were created by distant hybridization. So far, there are only few reports on the development of wheatTh. ponticum translocation lines resistant to this disease. Experimental methods used in this study (cytogenetic analyses-GISH and FISH; molecular marker analysis-Intro targeting marker amplification) were chosen properly and allowed to obtain the results which indicated the resistance genes of SN21171 and SN52684 were all derived from Th. ponticum. Besides, they suggested that stripe rust resistance of both translocation lines originates from novel genes derived from Th. ponticum. So that, these lines will contribute to wheat breeding resistance. References are appropriate and conclusions are consistent with the evidence and arguments presented in this research. Regarding the significant scientific value of this study for wheat breeding, I recommend to publish this manuscript in Plants (some editorial errors noticed in the text were already advisable in the previous revision).

Answer: Thanks for your opinion. We have carefully checked the paper again for English editing and the changes to the manuscript were also highlighted in yellow text.

Reviewer 2 Report

Comments and Suggestions for Authors

The study is of obvious interest for wheat breeding.

Below are some remarks.

References should be given as numbers in square brackets in the order they appear in the text.

Line 104:modify IT abbreveation for infection type because IT is used for IT markers.

Lines 84, 137: Intro targeting .> intron targeting

Line 84: IT marker amplification – add reference Choi et al (2004). A sequence-based genetic map of Medicago truncatula and comparison of marker colinearity with M. sativa. Genetics, 166(3), 1463-1502.

Line 128: “denatured at 105°C” – above the water nboiling point?

Section 2.4: It is necessary to describe the method for constructing intron-targeted markers and provide data on the primers used in IT marker analyses.

There is no reference to Table S1 in the text. Five specific IT markers of SN21171 and SN52684 not mentioned in the text.

Line 216: “2021-2022 years” – add references

Line 225: pools > pool, or delete an article a.

Author Response

Dear editor:

Thank you very much for sending us the reviewers’ comments on our manuscript (Plants-3294401) entitled “Cytogenetic identification and molecular marker analysis of two wheat-Thinopyrum ponticum translocations with stripe rust resistance”. Those comments are all valuable and helpful for revising and improving our paper. The following is the answers and revisions we have made in response to the reviewers’ questions and suggestions on an item by item basis. We have checked the paper again for English editing and the changes to the manuscript were also highlighted in yellow text.

Thanks again for your consideration of our manuscript. We are willing to respond to any further questions and comments that you may have.

Yours sincerely

Guotang Yang & Yinguang Bao

In response to the reviewers’ comments: (reviewers’ comments are marked in bold)

  1. The study is of obvious interest for wheat breeding. Below are some remarks.

References should be given as numbers in square brackets in the order they appear in the text.

Answer: Thanks for your suggestions. We have revised accordingly.

  1. Line 104: modify IT abbreveation for infection type because IT is used for IT markers.

Answer: Because IT abbreviation for infection type is widely accepted, intro targeting marker is not abbreviate as IT marker. All IT marker in this paper have been replaced with intro targeting marker.

  1. Lines 84, 137: Intro targeting > intron targeting

Answer: Revised accordingly. All intro targeting was replaced with intron targeting.

  1. Line 84: IT marker amplification – add reference Choi et al (2004). A sequence-based genetic map of Medicago truncatula and comparison of marker colinearity with M. sativa. Genetics, 166(3), 1463-1502.

Answer: Because the intro targeting marker amplification and primer data were provided by Zhang et al. (2017), the related reference was added (lines 139, 398-400).

  1. Zhang XD, Wei X, Xiao J, Yuan CX, Wu YF, Cao AZ, Xing LP, Chen PD, Zhang SZ, Wang XE, Wang HY (2017) Whole genome development of intron targeting (IT) markers specific for Dasypyrum villosum chromosomes based on next-generation se-quencing technology. Mol. Breed 37: 1–11. https://doi.org/10.1007/s11032-017-0710-0

  1. Line 128: “denatured at 105°C” – above the water nboiling point?

Answer: We have sorry for this spelling error. We have revised the 105°C as 100°C (line 126).

  1. Section 2.4: It is necessary to describe the method for constructing intron-targeted markers and provide data on the primers used in IT marker analyses.

Answer: These intro targeting markers have been constructed according to the study of Zhang et al. (2017). Related primers data also provided in this paper (lines 139, 398-400).

  1. There is no reference to Table S1 in the text. Five specific IT markers of SN21171 and SN52684 not mentioned in the text.

Answer: We added the Table S1 in line 202. Furthermore, 3 of 30 intron targeting markers could amplify the same bands in Th. ponticum, Th. bessarabicum and SN21171 genomes, but did not in Th. elongatum and Pseudoroegneria genomes (Fig. 4a, 4b, 4c, Table S1).

  1. Line 216: “2021-2022 years” – add references

Answer: We investigated the agronomic traits of two lines in 2021-2022 years. Thus, no reference was used in this place.

  1. Line 225: pools > pool, or delete an article a.

Answer: Revised accordingly. Th. ponticum, as a tertiary pool of common wheat, has many valuable genes useful for wheat genetic improvement. (line 224).

Reviewer 3 Report

Comments and Suggestions for Authors

ms titled "Cytogenetic Identification and Molecular Marker Analysis…." presents interesting results on wheat–Thinopyrum ponticum translocation lines. However, it has significant weaknesses that warrant rejection in its current form. Below are detailed comments outlining the critical issues:

·        The study does not present substantial innovation beyond existing reports on wheat–Thinopyrum ponticum translocation lines. Many similar studies have already explored resistance traits and agronomic evaluations in wheat lines derived from wild relatives.

·        Introduction is overloaded with refs, many tangentially related to the core study. This detracts from the focus and novelty of the manuscript.

·        Insufficient statistical analysis is reported for agronomic traits. There is no clarity on whether results are statistically significant, nor is there a discussion of variability between replicates.

·        The cytogenetic methodology section is overly descriptive, with unnecessary technical details that reduce readability without adding value.

·        The Discussion fails to adequately compare the findings with other wheat–Thinopyrum ponticum studies. It does not highlight how this work improves upon or differs from prior studies.

·        The manuscript suffers from redundancy and grammatical errors, especially in the Introduction and Discussion sections. This significantly impacts the readability and professionalism of the paper.

·        The Conclusion does not provide actionable insights or concrete directions for future work. It is too generalized to leave a lasting impression.

·        I noticed that your MS and the recently published article titled "Cytogenetic Identification and Molecular Marker Development for the Novel Stripe Rust-Resistant Wheat–Thinopyrum intermedium Translocation Line WTT11" share similar methodologies and objectives. While your study focuses on the wheat-T. ponticum translocation lines SN21171 and SN52684, the published article examines the WTT11 line derived from T. intermedium. To ensure clarity and strengthen the impact of your work, I recommend citing this article and briefly explaining how your findings differ. This will demonstrate awareness of relevant research, highlight the novelty of your study, and avoid criticism for overlooking prior work.

Comments on the Quality of English Language

The English could be improved to more clearly express the research.

Author Response

Dear editor:

Thank you very much for sending us the reviewers’ comments on our manuscript (Plants-3294401) entitled “Cytogenetic identification and molecular marker analysis of two wheat-Thinopyrum ponticum translocations with stripe rust resistance”. Those comments are all valuable and helpful for revising and improving our paper. The following is the answers and revisions we have made in response to the reviewers’ questions and suggestions on an item by item basis. We have checked the paper again for English editing and the changes to the manuscript were also highlighted in yellow text.

Thanks again for your consideration of our manuscript. We are willing to respond to any further questions and comments that you may have.

Yours sincerely

Guotang Yang & Yinguang Bao

In response to the reviewers’ comments: (reviewers’ comments are marked in bold)

  1. MS titled "Cytogenetic Identification and Molecular Marker Analysis…." presents interesting results on wheat–Thinopyrum ponticum translocation lines. However, it has significant weaknesses that warrant rejection in its current form. Below are detailed comments outlining the critical issues:·The study does not present substantial innovation beyond existing reports on wheat–Thinopyrum ponticum translocation lines. Many similar studies have already explored resistance traits and agronomic evaluations in wheat lines derived from wild relatives.

Answer: As we all know, the wheat–Thinopyrum ponticum translocation line is difficult to obtain because it is undergo the three difficult problems: cross incompatibility, hybrid incapacitation and segregation offspring. Furthermore, it is hardly reported that the wheat–Thinopyrum ponticum translocation line carry resistance to mstripe rust. Therefore, we considered that the development of new germplasm with stripe rust resistance is innovation in this paper.

  1. Introduction is overloaded with refs, many tangentially related to the core study. This detracts from the focus and novelty of the manuscript.

Answer: The structure and content of paper have been adjusted and modified.

  1. Insufficient statistical analysis is reported for agronomic traits. There is no clarity on whether results are statistically significant, nor is there a discussion of variability between replicates.

Answer: Because of some natural and man-made causes, the agronomic traits of two wheat–Thinopyrum ponticum translocation lines was investigated only in one season. In view of these good stripe rust resistance, the yield potential of two lines are currently being determined in field traits.

  1. The cytogenetic methodology section is overly descriptive, with unnecessary technical details that reduce readability without adding value.

Answer: In most of paper of this type, cytogenetic methodology is important section and described fully because of different operation in different scientific research team. To avoid the unnecessary technical details, some description of normal operation was been deleted.

  1. The Discussion fails to adequately compare the findings with other wheat–Thinopyrum ponticum studies. It does not highlight how this work improves upon or differs from prior studies.

Answer: The innovation in this paper is the development of two wheat–Thinopyrum ponticum translocation lines with stripe rust resistance. Furthermore, the chromosome constitution and source of alien chromosomal segments of two translocation lines were different from other prior studies. Therefore, more differs from prior studies were added and compared.

  1. The manuscript suffers from redundancy and grammatical errors, especially in the Introduction and Discussion sections. This significantly impacts the readability and professionalism of the paper.

Answer: The structure, content and grammatical errors of paper have been adjusted and modified.

  1. The Conclusion does not provide actionable insights or concrete directions for future work. It is too generalized to leave a lasting impression.

Answer: The conclusion was modified accordingly.

  1. I noticed that your MS and the recently published article titled "Cytogenetic Identification and Molecular Marker Development for the Novel Stripe Rust-Resistant Wheat–Thinopyrum intermedium Translocation Line WTT11" share similar methodologies and objectives. While your study focuses on the wheat-T. ponticum translocation lines SN21171 and SN52684, the published article examines the WTT11 line derived from T. intermedium. To ensure clarity and strengthen the impact of your work, I recommend citing this article and briefly explaining how your findings differ. This will demonstrate awareness of relevant research, highlight the novelty of your study, and avoid criticism for overlooking prior work.

Answer: The two studies are different completely.

(1) These translocation lines have different sources. WTT11 is the offspring of wheat and Th. intermedium. SN21171 and SN52684 are the offspring of wheat and Th. ponticum.

(2) Their translocation chromosome constitution is different. WTT11 was underwent the T7JsL·2DL translocation and SN21171 and SN52684 separately carry T3Eb-3DS·3DL and T1Eb-1BS·1BL translocation. The source of alien chromosomal segments is different.

(3) The structural variations in wheat chromosomes were not identified in WTT11, whereas the different FISH signals were compared in SN21171 and SN52684 and their wheat recurrent parent.

Round 2

Reviewer 3 Report

Comments and Suggestions for Authors

it can be accepted in this current form